# Moral conformity in a digital world: Human and nonhuman agents as a source of social pressure for judgments of moral character

**Konrad Bocian** [1]*, **Lazaros Gonidis**[2], **Jim A.C. Everett**[3]

**1** Department of Psychology in Sopot, SWPS University, Warszawa, Poland, **2** School of Psychology, University of Sussex, Brighton, United Kingdom, **3** School of Psychology, University of Kent, Canterbury, United Kingdom

* kbocian1@swps.edu.pl

## Abstract

Could judgments about others' moral character be changed under group pressure produced by human and virtual agents? In Study 1 ($N = 103$), participants first judged targets' moral character privately and two weeks later in the presence of real humans. Analysis of how many times participants changed their private moral judgments under group pressure showed that moral conformity occurred, on average, 43% of the time. In Study 2 ($N = 138$), we extended this using Virtual Reality, where group pressure was produced either by avatars allegedly controlled by humans or AI. While replicating the effect of moral conformity (at 28% of the time), we find that the moral conformity for the human and AI-controlled avatars did not differ. Our results suggest that human and nonhuman groups shape moral character judgments in both the physical and virtual worlds, shedding new light on the potential social consequences of moral conformity in the modern digital world.

## Introduction

For six decades, psychology undergraduates have memorised details of Asch's [1] landmark study on conformity, learning that a significant number of participants conformed by giving an incorrect response in a perceptual discrimination task when responding after a series of confederates who gave a different answer. But what if participants are not judging how long a series of lines are but something much more consequential–their moral beliefs about right and wrong? And what if, in our increasingly digital world, the confederates were not physical human confederates but artificial agents in a virtual space? Bringing together work on the independent importance of social conformity [2], moral character judgments [3], and social influence in digital and immersive virtual environments [4], in this paper, we answer these questions, investigating how human and nonhuman sources of group pressure shape perceptions of moral character.

**Data Availability Statement:** All raw data files, analysis scripts, and materials used in this article

are available for download from the Open Science Framework: https://osf.io/aq4tg/.

**Funding:** "The preparation of this paper was supported by the Polish National Science Centre grant 2018/02/X/HS6/02164 (MINIATURA 2) and by the European Association of Social Psychology Seedcorn grant awarded to Konrad Bocian. The funders had no role in study design, data collection and analysis, decision to publish, or preparation of the manuscript".

**Competing interests:** The authors declare that there are no potential conflicts of interest with respect to the research, authorship, and/or publication of this article.

## Social conformity

Social conformity is changing a belief or behaviour to match the responses of the majority [5]. According to Deutsch and Gerrard [6] (1955), people conform under the pressure of others for two reasons. First, they want to fit in with the group and obtain social approval (normative conformity). Second, they lack sufficient knowledge and view the group as a source that accurately interprets the current situation (informational conformity). In his famous study, Asch [1] showed that people were willing to accept the group's incorrect answer although they knew the correct answer: people complied with the group publicly but disagreed with it privately. Therefore, we have evidence that people conform if group pressure is present but stop complying when group pressure fades away. For over 50 years, numerous studies have used the classic Asch paradigm to examine social conformity's influence on various judgments, including line or colour judgments, judgments about intelligence or just opinions [7]. Nevertheless, there are situations where conformity is likely to be much more impactful than answers to questions about lines and colours. One area where conformity is likely to have serious social consequences is potential conformity in the moral domain.

## Moral conformity

Considering robust evidence confirming the importance of morality in everyday life [see 8], it is surprising that only a few studies have investigated whether moral judgments are subject to group pressure. For example, Kundu and Cummins [9] asked participants to decide about moral dilemmas privately or in a group of confederates. They found that both permissible and impermissible actions were influenced by group pressure. Hence, participants' judgments of permissibility aligned with the judgments of the confederates regarding the permissibility of immoral and moral actions [9].

Similar results were found in a study where participants judged violations of moral, social and decency norms in the presence of social pressure. This study showed that people mainly conformed while judging violations of social and decency norms and least for moral norms [10]. Moreover, such moral conformity emerges early, with evidence of conformity among pre-schoolers who changed their social and moral judgments under social pressure [11]. Further, there is evidence that moral conformity might be sensitive to what *kind* of moral judgement people make, with people conforming to a deontological but not a consequentialist majority [12], in line with evidence for the negative reputational costs of making consequentialist decisions [13, 14]. Finally, during a video meeting, individuals tend to conform with the group when faced with sacrificial moral dilemmas [15].

The relatively scant evidence on *moral* conformity suggests that social influence shapes participants' moral judgments. However, previous work on moral conformity has looked only at ratings of the moral wrongness of an action. Yet, there is an increasing consensus that judgments about someone's moral character are as, if not *more*, central to our moral psychology. According to the person-centred approach to morality [16, 17], when faced with moral violations, people are not necessarily asking whether a given action is right or wrong but whether the *person* who did the action is good or bad.

This is perhaps unsurprising, considering data showing that a person's perception is underlain by two content dimensions–morality and competence [18]. Perceptions of morality and competence account for 82% of the variance in global impressions of people [19]. More recent research corroborates this data, showing that the moral dimension is central to a person's perception process [20]. Moreover, impressions about others are substantially changed when moral information but not about sociability or competence is added [21]. Finally, the recently proposed Moral Primacy Model (MPM) of impression development shows that moral

information dominates each stage of impression formation: gathering information, making first impressions, and revising the impression [22].

Perceptions of moral character have a host of social consequences, shaping character inferences, trusting behaviour in economic games, and perceived suitability for different social roles [23]. Further, they determine whom people approach and avoid [24], and in extreme cases, they shape life or death decisions [25]. Therefore, and in contrast to previous studies, we aimed to investigate to what extent private moral character judgments could be changed under the pressure of a group.

Based on the evidence from research on social and moral conformity, we predicted that private moral character judgments would be impacted by group pressure. Specifically, we assumed that in public, participants' moral character judgments would align with the confederates' judgments and, therefore, differ from their private moral character judgments (Hypothesis 1). To this end, we recorded how many times participants changed their private moral character judgments under the pressure of the group.

## Do people conform in immersive virtual environments (IVE)?

Almost 70 years ago, Gordon Allport defined social psychology as an attempt to understand and explain how others' actual, imagined or implied presence influences our thoughts, feelings, and behaviour [26]. Twenty years later, Blascovich suggested that social influence should also occur in digital and immersive virtual environments in the presence of *virtual* others [27]. According to the Threshold Model of Social Influence (TMSI; [4]), social presence positively impacts social influence. The TMSI assumes that the more individuals perceive themselves within interpersonal or social environments, the greater the social influence. However, social presence varies as a function of several factors. Two of them are agency and behavioural realism.

In the TMSI, the agency is defined as the extent to which individuals perceive virtual others as representing real persons. Therefore, the agency is represented as a continuum, anchored on the low end, where agents are perceived as entirely controlled by non-human means (e.g., cyborgs, autonomous vehicles). On the high end, we have agents completely controlled by real humans (e.g., avatars., drones). Behavioural realism, in turn, refers to the degree to which virtual objects act as they would in the physical world. Similar to the agency, it is represented on the continuum from the low to the high end. The TMSI assumes that social presence should increase if the agency is high. Social presence should also increase if behavioural realism is [4].

Today, we observe a significant development of immersive virtual technology in the modern digital world. The metaverse is a universal and immersive virtual world which can be experienced using virtual reality (VR) and augmented reality (AR) headsets [28]. Metaverses are on track to become increasingly important in the social world, with companies like Facebook explicitly moving towards a metaverse framework [29]. We are increasingly facing a world in which social influence may be equally, if not more, potent in the digital world as it is in the real world. Therefore, we argue that social and moral psychology needs to know how the processes and manifestation of moral conformity might be different in our modern digital age, in which social interactions are mediated through remote, online technology (e.g., social media, metaverses) - and some of the critical social observers are not even human at all.

Evidence of social conformity in metaverse-style immersive virtual environments (IVEs) is relatively scant. One study has shown that people could be influenced by avatars representing other people [30]. Specifically, participants played at a blackjack table in a digital immersive virtual casino. In the first round, participants played alone, while in the second round, with two other players described as agents controlled by non-humans or avatars controlled online

by real humans. The role of the other players was to manipulate the betting norms, so as a result, participants would systematically bet more. Participants' betting averages confirmed that participants conformed to the behaviour of other players. Interestingly, the conformity levels did not depend on whether players were controlled by non-human agents or real humans [30].

In one particularly relevant recent study, participants participated in a traditional Asch paradigm represented in virtual reality. Like the original experiment, participants saw boards with three lines and had to compare the lines to the reference line. Before participants answered, five other avatars gave incorrect answers. Additionally, participants were told that people in other labs or computers controlled avatars via algorithms. The results showed that virtual humans could produce social conformity in immersive virtual environments. However, levels of conformity did not depend on whether avatars were controlled by humans or computers [31].

Other evidence regarding social conformity is not related to immersive virtual reality but confirms that people conform under the pressure of non-human agents. For example, immersive video gaming increases conformity to judgments cast by artificial intelligence, especially when the stimulus context is ambiguous [32]. Moreover, in impersonal digital settings, social information (e.g., the frequency of specific responses) shapes moral judgments [33] and reduces verbal aggression [34]. Finally, past work demonstrated that humans do not conform to the presence of robots [35], but later evidence confirmed that they do [36].

Although limited, evidence suggests that people conform to the group pressure of avatars representing humans in the immersive virtual environment. However, we still need to find out whether people would conform and change their moral character judgments in such environments. Based on the assumptions of the TMSI model [4] and that we did not manipulate the agency and behavioural realism of the avatars, we may assume that people would change their moral character judgments to the same extent under the pressure of avatars allegedly controlled by other humans or artificial intelligence. However, since evidence suggests that people sometimes conform to machines, especially ones who pretend to be humans [see 37], we may also assume that people would conform more to avatars controlled by other humans than AI (Hypothesis 2).

## What factors may impact moral conformity?

People differ in their judgments of what acts are right or wrong. According to the influential Moral Foundations Theory [38], the moral domain can be demarcated into people's concerns about harm/care, fairness, loyalty, authority, and sanctity. Significantly, political orientation influences people's moral judgments across these foundations. For example, when judging others' behaviour as moral or immoral, liberals consistently show more significant endorsement and use of harm and fairness and less loyalty, authority, and purity foundations. In contrast, conservatives endorsed and used all five foundations to a more similar extent [39].

In other words, some acts (e.g., kicking a dog) will be broadly equally wrong regardless of political orientation because they violate the harm foundation, which liberals and conservatives endorse to a similar extent. However, a different act (e.g., disloyalty through flag-burning) would be judged as significantly more wrong by conservatives, but not liberals, because they focus on the loyalty foundation when judging whether something is right or wrong [39].

This suggests that people's compliance under group pressure may depend on their political orientation. Therefore, we explored whether liberals would conform less when moral judgments concern targets who harm or deceive others because care and fairness foundations are fundamental for their moral reasoning. We also explored if conservatives would conform less when moral judgments concern targets who showed disloyalty, disrespect for authority, or

violated purity standards, as loyalty, authority, and purity foundations are critical for their moral reasoning (Hypothesis 3).

In contrast to the MFT, the Theory of Dyadic Morality [40] argues that all moral foundations specified by the MFT have a common denominator: harm. In other words, the Theory of Dyadic Morality (TDM) argues that moral cognition is rooted in a harm-based template because harm is central to moral cognition. Therefore, when fairness, loyalty, authority, or purity foundations are violated, people, in the first place, judge if and what kind of harm was done. Indeed, evidence suggests that harm is the most accessible and essential moral content for both liberals and conservatives [41]. This suggests that foundations proposed by the MFT are not related to specific moral content (e.g., loyalty, purity) but resemble different perceptions of harm [40].

Based on the Theory of Dyadic Morality, gathered evidence and the assumption that perception of harm is central to moral cognition, we explored if participants, independently of their political orientation, would change their moral judgments less when these judgments concern targets who harm others (Hypothesis 4).

## Study 1

In the first study, we aimed to investigate to what extent participants' moral character judgments would change under the pressure of real humans. To this end, we measured participants' moral character judgments first privately and two weeks later in the presence of three confederates who made opposite moral character judgments to participants, investigating whether those participants would change their private moral character judgments under group pressure to align them with the moral character judgments of the majority.

### Method

In this article, we report all measures and any data exclusions. Any additional measures not included in the primary analyses are reported in the Supplement. The reported studies were approved by the ethical committee of SWPS University (Ethics Clearance ID: WKE/S 4/X/62) and the University of Kent (Ethics Clearance ID: 201915695053005867). All participants provided informed consent. All raw data files, analysis scripts, and materials used in this article are available for download from the Open Science Framework: https://osf.io/aq4tg/. Studies in this manuscript were initially preregistered as one study based on the small grant received by the authors. However, due to helpful comments from reviewers, we realised that the pre-registration needed to be revised. In the end, we split the studies into two without further preregistration.

**Design.** We modified the Asch [1] conformity paradigm in several ways to test our predictions. In contrast to the Asch [1] conformity paradigm, before our participants arrived at the lab, they judged privately at home whether the agent of the behaviour presented in a vignette was a good or bad person. After no later than two weeks, we asked participants to make the same moral character judgments publicly in the presence of three peers. As we knew the participants' answers from the first part of the study, the group always provided answers opposite to the participants' responses. Participants were in a position where they had to answer as the last person in their group. Thus, a key-dependent measure was how many times out of the 20 responses, participants changed their moral character judgments under the group's influence.

We expected that participants under group pressure would change some of their private judgments about the target's moral character. Moreover, we tested, on the one hand, if political orientation would impact levels of conformity dependently on a violated moral foundation.

On the other hand, if people, independent of their political orientation, would change their moral character judgement less if their concern targets who harm others.

**Participants.**   We did not use power analysis for sample size estimation when planning the study. Instead, we used a rule of thumb and aimed to recruit at least 50 participants [42]. Ultimately, we recruited 103 participants from a Polish university pooling sample (92 women; $M_{age}$ = 22.13 years, $SD_{age}$ = 7.05) who participated in the study in exchange for course credit. The recruitment period started on 2$^{nd}$ October 2019 and ended on 20$^{th}$ November 2019.

**Procedure.**   At least two weeks before participants arrived at the laboratory, we sent them the online questionnaire link. We explained that in the first part, we would ask them to read 20 different vignettes online and judge the target person presented in each vignette. Further, we informed participants that when they completed the first part of the study, we would invite them to the second part, which would be conducted in the laboratory. Vignettes presented to participants described short behaviours violating either one of five moral foundations (care, fairness, loyalty, authority, sanctity) or a non-moral social norm. In both parts of the study, and for each condition, vignettes were presented to participants in random order. We used the non-moral violations because past studies showed that people conformed more to social than moral norms [10]. We used moral foundations vignettes validated by Clifford et al. [43] because each vignette depicts a behaviour violating a particular moral foundation and not others. We present some examples of foundation violations below (see the Supplement for the full text for each vignette):

Care Foundation

*You see a woman spanking her child with a spatula for getting bad grades in school.*

Fairness Foundation

*You see a woman getting hired only because her father is close friends with the boss.*

Authority Foundation

*You see a teaching assistant talking back to the teacher in front of the classroom.*

Loyalty Foundation

*You see a head cheerleader booing her high school's team during a homecoming game.*

Sanctity Foundation

*You see two first cousins getting married to each other in an elaborate wedding.*

Social Norms

*You see a man making a phone call in a cinema and talking loudly.*

Based on the respondents' classifications, we selected behaviours where the mean wrongness was estimated at 2.5 on the 5-point scale to present not extreme violations but somewhat ambivalent in their character. Therefore, for each moral foundation and non-moral social norm, we randomly presented four different violations to participants. We found only two behaviours violating the Authority foundation and two violating the Loyalty foundation,

which wrongness was estimated at 2.5 on the 5-point scale. Therefore, these two foundations were represented by two vignettes instead of four. After reading the vignette, participants had to answer the question: "Do you think that [TARGET] is mainly a good person or a bad person?" with either the option "Mainly a good person" or "Mainly a bad person".

Two weeks later, participants came to the laboratory for the second part, where the other three confederates pretended to wait for their turn to participate in the study (see the Supplement for the setting's picture). After a minute or less, an experimenter showed up and explained that she would like to test everyone simultaneously because she was running late. After getting verbal approval from each participant to be tested in the group, confederates and participants were invited to another room. In the room, there was a table and four chairs. Participants always sat on the first chair from the door. Still, they answered last because the experimenter indicated that answers would be given in order, starting from the person sitting farthest from the door.

The experimenter explained to the participants that she would like to present the vignettes from the first part and ask about their opinions again. The vignettes were presented in random order using a wall projector. The experimenter read each vignette aloud and then asked each person in the room whether the target person was mainly a good or a bad person. Confederates always answered the opposite of the participants' answers in the first part. For example, if a participant's answer in the first stage was: "Mainly a good person", in the second stage, the confederates' answer was: "Mainly a bad person".

## Measures

**Moral conformity ratio** was measured by counting how many times out of 20 participants changed their moral judgments (from good to bad or from bad to good) about the target under group pressures compared to their initial private moral judgments.

**Judgment confidence** was measured to control participants' confidence in their answers. Thus, participants read five statements: "The answers I gave in the test were correct", "I have doubts about the correctness of the answers I gave" (reversed-scored), "The answers given by the other participants affected my own answers", (reversed-scored), "The answers I gave were mainly based on my own opinion" and "I felt confident about my answers" and indicated to what extent they agree with each of the statement using a scale from 1 = *strongly disagree* to 7 = *strongly agree* with higher ratings showing higher confidence ($\alpha = 0.75$, $M = 5.26$, $SD = 1.02$).

**Political ideology** was measured with a single item to test whether political orientation would moderate conformity depending on which moral foundations were violated. Participants were asked to report their political ideology on a scale from 1 = *extremely liberal* to 8 = *extremely conservative* ($M = 3.37$, $SD = 1.86$).

## Results

**Private and public judgments.** To test whether participants changed their private moral character judgments to align them with judgments of confederates (H1), we ran McNemar's test, which pairs nominal data. Therefore, we paired and compared frequencies of moral character judgments that participants made privately and later publicly under group pressure for each vignette and condition separately. This analysis confirmed that participants' moral character judgments aligned with the confederates' judgments. Under the pressure of actual humans, participants changed their private judgments to 15 vignettes out of 20. Overall, participants aligned their moral character judgments with real humans' judgments 43% of the time (See Table 1).

**Table 1. Number of participants who changed and did not change their private moral character judgments under group pressure in studies 1 & 2.**

| Foundation/Vignette | | Study (Source of pressure) | | | | | | | |
|---|---|---|---|---|---|---|---|---|---|
| | | Study 1 (Human) | | | Study 2 (Human Avatar) | | | Study 2 (AI Avatar) | | |
| | | Changed Private Judgment | | $p$ | Changed Private Judgment | | $p$ | Changed Private Judgment | | $p$ |
| | | Yes | No | | Yes | No | | Yes | No | |
| Care | Laugh | 32 | 71 | < .001 | 27 | 43 | < .001 | 20 | 48 | .012 |
| | Dinner | 29 | 74 | .024 | 21 | 49 | .027 | 13 | 55 | .267 |
| | Spatula | 42 | 60 | .001 | 11 | 59 | .227 | 13 | 55 | .267 |
| | Dog | 29 | 74 | .008 | 19 | 51 | .004 | 15 | 53 | .118 |
| Fairness | Player | 53 | 49 | .002 | 22 | 48 | .004 | 20 | 48 | .003 |
| | Halloween | 53 | 49 | < .001 | 32 | 38 | < .001 | 25 | 43 | < .001 |
| | Line | 64 | 38 | .005 | 30 | 40 | < .001 | 26 | 42 | .029 |
| | Hired | 43 | 59 | .001 | 13 | 57 | .006 | 12 | 56 | .012 |
| Authority | Order | 49 | 53 | .021 | 26 | 44 | .076 | 25 | 43 | .108 |
| | Teacher | 50 | 53 | < .001 | 29 | 41 | .265 | 21 | 47 | .003 |
| Loyalty | General | 43 | 60 | .032 | 28 | 42 | .215 | 30 | 38 | .856 |
| | Cheerleader | 56 | 47 | .894 | 24 | 46 | .307 | 16 | 52 | .004 |
| Sanctity | Toothbrush | 52 | 50 | .070 | 18 | 52 | .002 | 19 | 49 | .004 |
| | Gay | 46 | 56 | < .001 | 17 | 53 | < .001 | 17 | 51 | < .001 |
| | Cousins | 44 | 57 | .291 | 19 | 51 | < .001 | 19 | 49 | < .001 |
| | Vomits | 45 | 57 | .079 | 15 | 55 | .006 | 14 | 54 | .025 |
| Social Norm | Phone | 60 | 43 | < .001 | 12 | 58 | .009 | 15 | 53 | .607 |
| | Gift | 29 | 74 | .442 | 20 | 50 | .094 | 15 | 53 | .263 |
| | Hello | 53 | 50 | .013 | 25 | 45 | .015 | 12 | 56 | < .001 |
| | Desert | 7 | 96 | .125 | 7 | 63 | .262 | 6 | 62 | .687 |
| Mean | | 43.95 | 58.5 | | 20.75 | 49.25 | | 17.65 | 50.35 | |
| SD | | 13.31 | 13.53 | | 6.93 | 6.93 | | 5.79 | 5.79 | |
| Mean per cent | | 43% | 57% | | 30% | 70% | | 26% | 74% | |

*Note.* The difference between frequency of private and public judgments was analyzed with McNemar's test. Value of *p* indicates whether this difference was significant.

**Moral foundations and political orientation.** To test whether the moral foundation in which the violation was committed would impact participants' conformity and their political orientation, we first estimated the conformity ratio (CR) for each participant. The CR was estimated with the formula: [x]/20 (number of judgements) = CR, where [x] indicates how many times participants changed their initial private judgments to align with judgments made by humans or avatars. Afterwards, we performed 2 (political orientation: liberal vs. conservative) x 6 (foundation: care vs. fairness vs. authority vs. loyalty vs. purity vs. social norm) mixed-model ANOVA with the first factor between and the second within participants. This analysis yielded a main effect of the foundation source, $F(4.417, 446.139) = 6.57$, $p < .001$, $\eta_p^2 = .06$, 95% CI [.02, .10], (see Fig 1).

Following pairwise comparisons, we found that participants conformed less when their judgements concern care ($M = 0.32$, $SD = 0.27$) than fairness ($M = 0.52$, $SD = 0.31$), $p < .001$, $d = -0.36$, 95% CI [-0.77, -0.36], obedience to authority ($M = 0.48$, $SD = 0.40$), $p < .001$, $d = -0.44$, 95% CI [-0.56, -0.17], loyalty ($M = 0.48$, $SD = 0.34$), $p < .001$, $d = -0.40$, 95% CI [-0.60, -0.20], and sanctity ($M = 0.46$, $SD = 0.31$), $p < .001$, $d = -0.39$, 95% CI [-0.52, -0.15] but to the same extent to violations of social norms ($M = 0.36$, $SD = 0.26$), $p = .108$, $d = -0.34$, 95% CI [-0.32, 0.07], (see the Supplement for remaining pairwise comparisons).

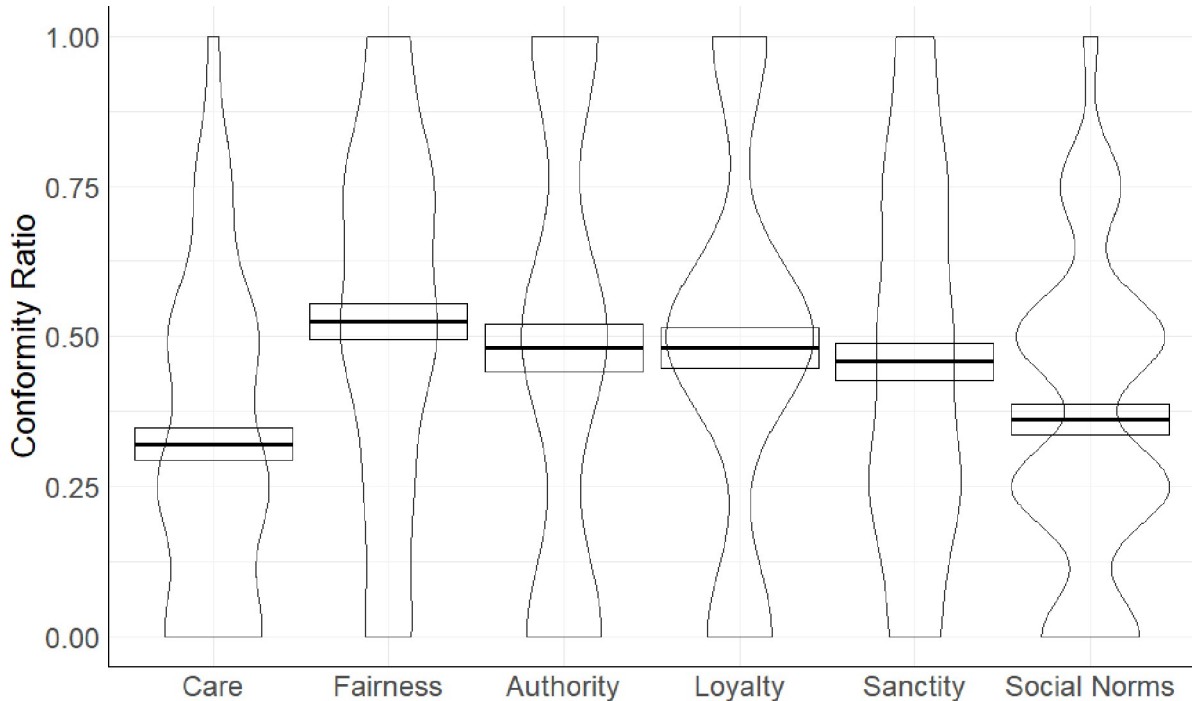

**Fig 1. Mean conformity ratio as a function of moral domains and social norms.** The thick black horizontal line represents the mean with one standard error.

The main effect of the political orientation was nonsignificant, $F(1, 101) = 0.17$, $p = .683$, and the interaction effect between political orientation and foundations, $F(5, 505) = 0.64$, $p = .772$. Therefore, these results confirm H4 because people conformed less when their judgments concerned targets who harm others. We did not find evidence for H3 as people's compliance under group pressure did not depend on their political orientation.

## Discussion

Study 1 confirmed our first hypothesis that people are willing to change their private moral character judgments under the group pressure of other humans. Results of Study 1 also provided evidence for our fourth hypothesis because compliance with the group was more minor when moral character judgments concerned targets who harm others in contrast to targets who broke moral norms of fairness, authority, loyalty, or purity. Finally, the third hypothesis was not confirmed because political orientation did not impact moral conformity in different moral domains.

## Study 2

In the second study, we sought to replicate the results of Study 1 in a different context: an immersive virtual environment. Therefore, we asked participants to make the same moral character judgments as in Study 1 in the presence of three avatars allegedly controlled by humans or AI. We assumed that participants would comply with the group of avatars and change their private moral character judgments. We also predicted that there would be no differences in their moral conformity between the AI and Human-controlled avatars conditions.

## Method

**Participants.** We planned to collect data from 200 participants (100 per condition) to allow for technical problems and potential exclusions. In the end, we recruited 138 participants (115 women; $M_{age}$ = 19.30 years, $SD_{age}$ = 1.57) from the British university pooling sample. The recruitment period started on 24th October 2019 and ended on 16th March 2020. Unfortunately, towards the end of our data collection, the COVID-19 pandemic began, and all data collection was halted. Therefore, to assess the power we obtained, we conducted a sensitivity power analysis revealing that our sample size provides a power of 0.80 to detect an effect size of $f^2$ = 0.09.

**Setting and environment.** The avatars and their head rotation animations were created in Blender version 2.7, and we then passed these as assets in Unity to design the rest of the experimental setting. We ran the experiment in Unity version 2019.2.4f1. We used a high-performance gaming laptop with an Intel i7-9750H processor, 16 GB of RAM and an Nvidia 2070 Max-Q graphics processor. Regarding the virtual reality headset, we have an Oculus Rift, with its sensor placed and calibrated at 1m from the participant's seat.

Two female and one female avatar were seated in front of a table, having their hands placed on the table. They are mainly static except for rotating their heads to face other avatars or the participant. This would happen at random between 30 and 120 seconds for a random period between 4 and 10 seconds (see Fig 2).

Avatars did not speak. Only the "virtual experiment" would speak, prompting each participant to submit a response. The response was not vocal. Instead, the text "good" or "bad" would appear in each "participant's" corresponding text box on the whiteboard (see Fig 3).

Participants were instructed to press the left or right trigger to respond "good" or "bad" using an Xbox controller. Once the "virtual research assistant" asked the participant to respond with either "good person" or "bad person," the participant would press the corresponding to their choice controller trigger (see Fig 4).

**Procedure.** We used the same procedure for the first part of the study as in Study 1. However, there were some differences in the second part. First, participants arrived individually at the laboratory, where the experimenter told them they came as the fourth and last person. In the human-controlled avatars setting, they were further told that the other three participants already connected the VR headsets and were waiting in separate rooms.

Moreover, because of the time constraints, the study was run as a group of four in VR so that they would answer questions in the order they arrived at the lab. In the AI-controlled avatars setting, participants were told the Kent School of Engineering and Digital Arts wanted to run tests on their new three algorithms, which were implemented in the virtual avatars. Because of this, the study is run as a group of four in VR, where the avatars would answer first, and the participant would answer last.

Participants sat alone in the room where the experimenter mounted the VR headset on their heads and briefly explained the digital environment and how they should use the controllers to answer the questions. In the IVE, participants saw three other avatars, the avatar of the experimenter and a whiteboard on the wall where the questions and avatars' responses were projected (see the Supplement for the overview of the digital setting). Like in Study 1, avatars always give their answers before participants. Furthermore, the avatars' answers were always the opposite of what participants had responded to in the first part. After the group part ended, participants were escorted to individual cubicles and asked to answer follow-up questions.

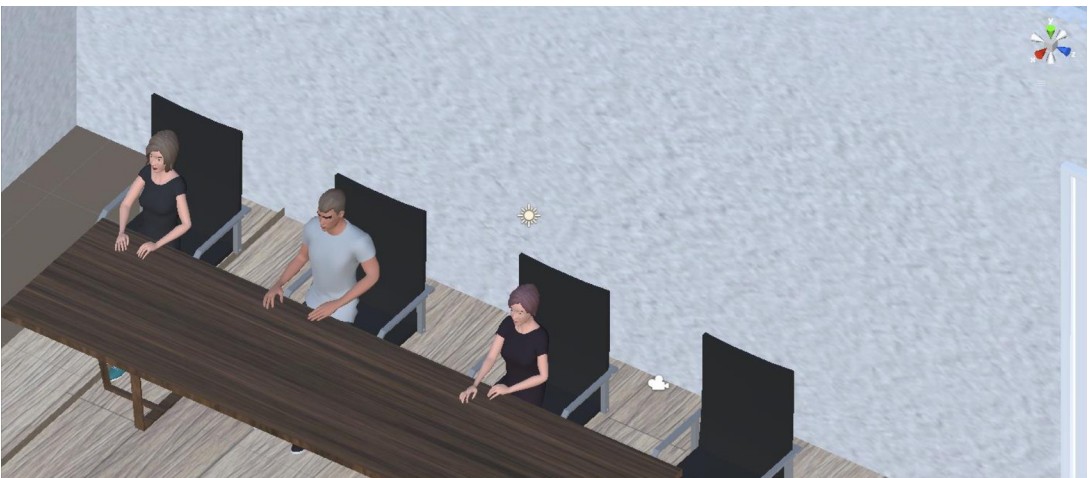

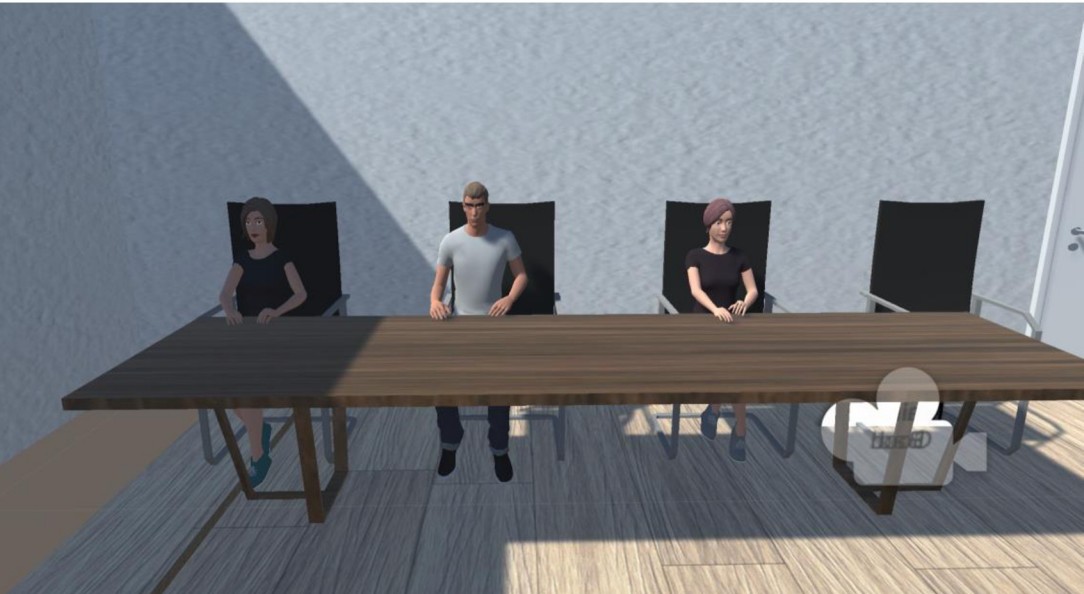

**Fig 2. The example of virtual peer pressure setting in Study 2.**

## Measures

**Moral conformity** was measured as in Study 1.

 **Judgment confidence** was measured as in Study 1 ($\alpha = 0.69$, $M = 5.50$, $SD = 0.91$).

 **Political ideology** was measured as in Study 1 ($M = 3.75$, $SD = 1.34$).

## Results

 **Private and public judgments.** We first tested if participants changed their private moral character judgments to align them with judgments of avatars. When avatars allegedly controlled by humans produced the group pressure, we observed a change for 13 vignettes out of 20. When AI allegedly controlled avatars produced group pressure, change considered 12 vignettes out of 20. Overall, participants aligned their moral character judgments with

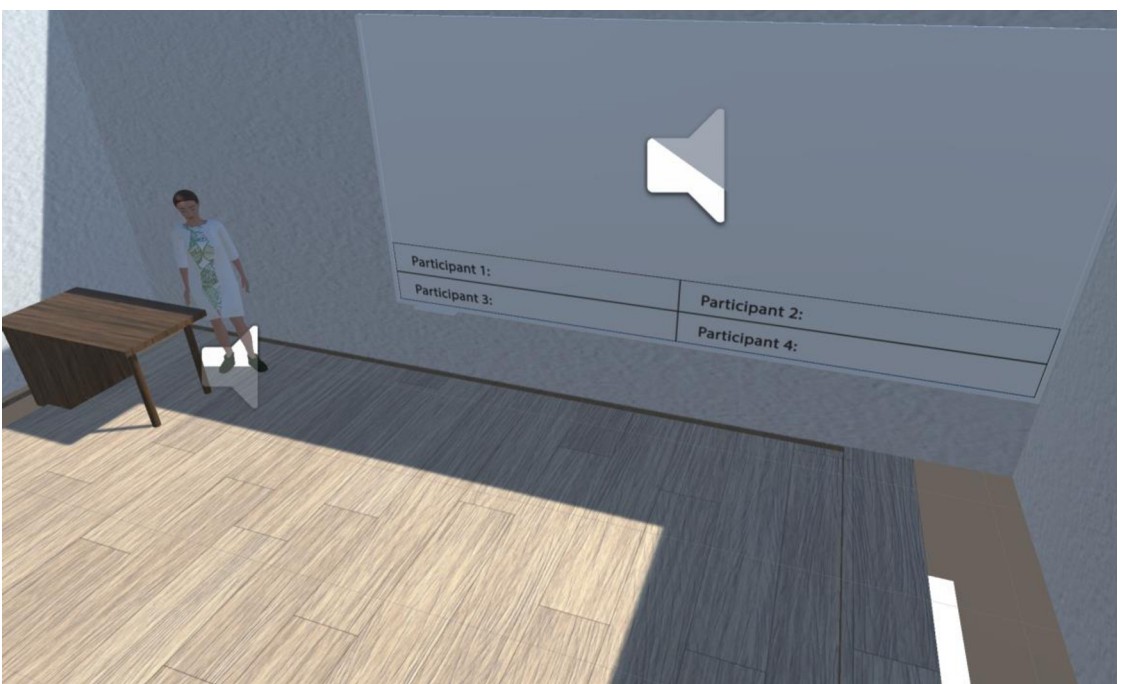

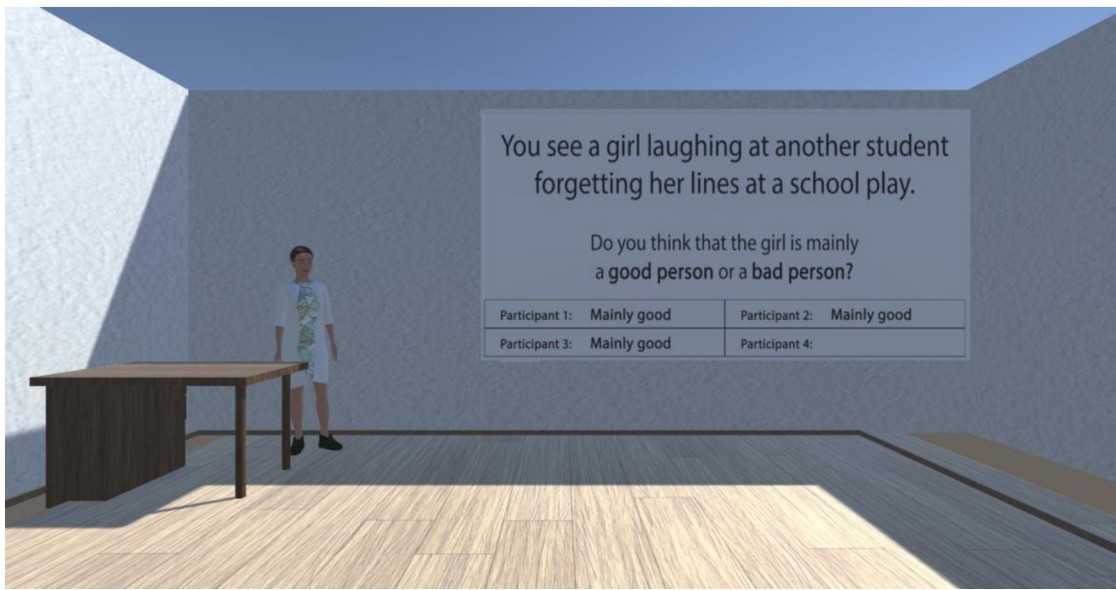

**Fig 3. The example of stimuli presentation in the virtual setting in Study 2.**

judgments of avatars controlled by humans 30% of the time and 26% when avatars were controlled by AI (See Table 1).

**Moral foundations, political orientation, and source of pressure.** We performed 2 (source of pressure: human avatars vs. AI avatars) x 2 (political orientation: liberal vs. conservative) x 6 (foundation: care vs. fairness vs. authority vs. loyalty vs. purity vs. social norm) mixed-model ANOVA with two first factors between and the third within participants.

The main effect of the source of pressure was nonsignificant, $F(1, 134) = 2.00$, $p = .160$, as well as the main effect of the political orientation, $F(1, 134) = 0.39$, $p = .533$. As predicted, we

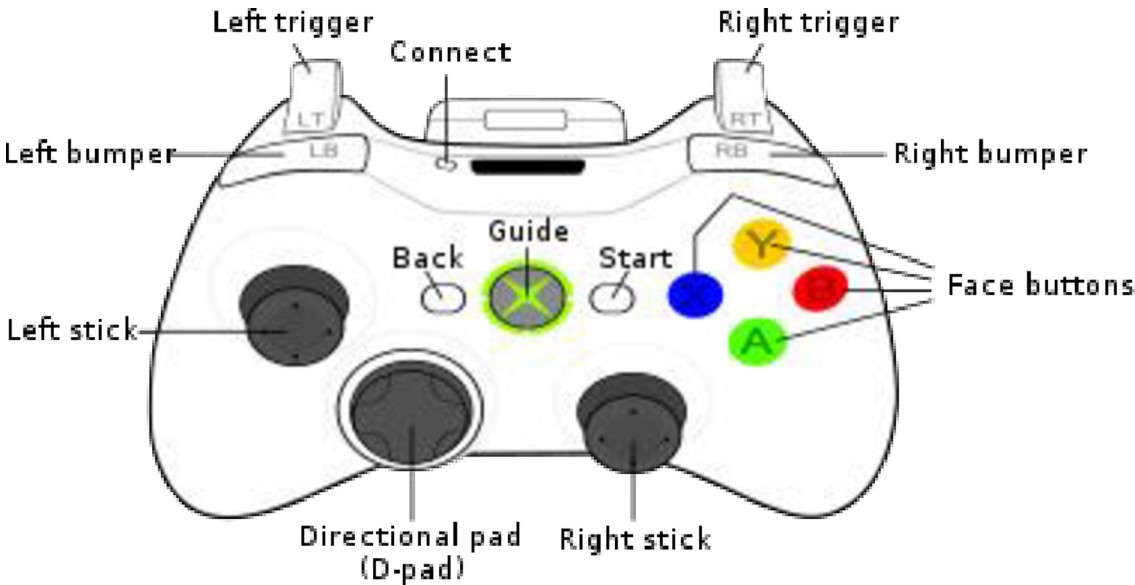

**Fig 4. The example of the controller and assigned triggers which participants used in the virtual setting in Study 2.**

found no difference between human-controlled and AI-controlled avatars conditions in participants' feelings of presence and realism in the IVE (see the Supplement).

However, we found the main effect of the foundation, $F(4.259, 570.748) = 6.89$, $p < .001$, $\eta_p^2 = .05$, 95% CI [.02, .08] (see Fig 5).

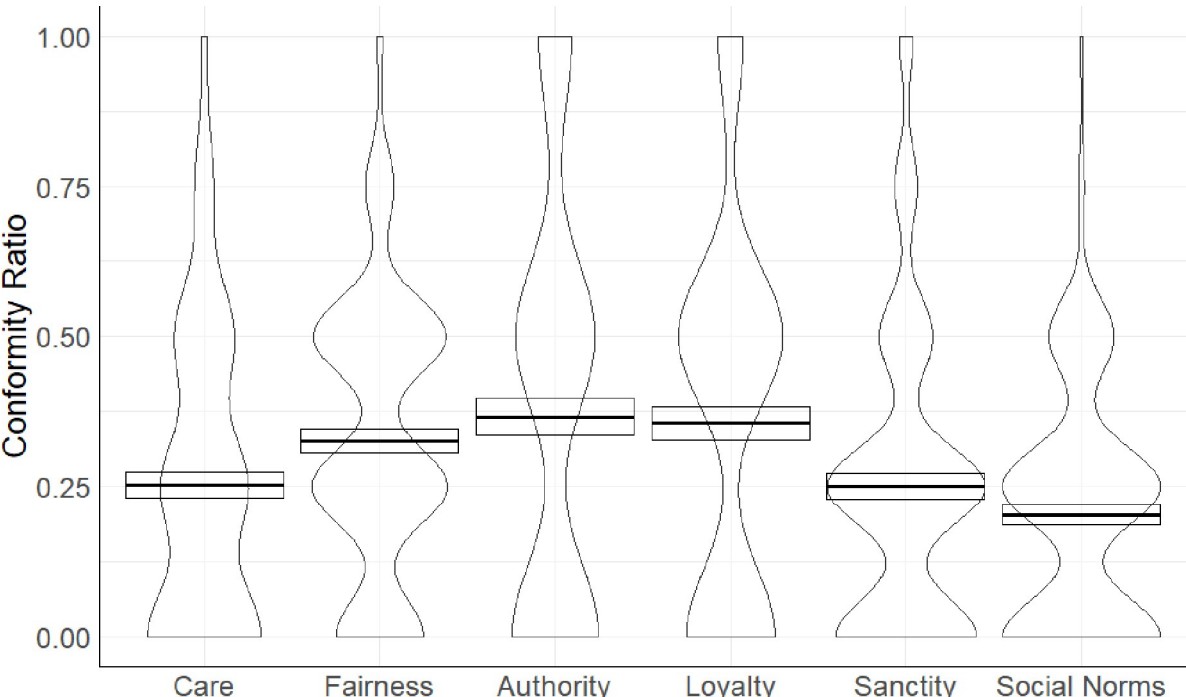

**Fig 5. Mean conformity ratio as a function of moral domains and social norms.** The thick black horizontal line represents the mean with one standard error.

In contrast to Study 1, using pairwise comparisons, we found that participants conformed less when their judgements concern care ($M$ = 0.25, $SD$ = 0.26) than authority ($M$ = 0.37, $SD$ = 0.37), $p < .001$, $d$ = -0.40, 95% CI [-0.45, -0.11]), but comparisons with other foundations were nonsignificant, $ps$ = .830, (see the Supplement for remaining pairwise comparisons).

Interestingly, we also found the three-way interaction effect between the source of pressure, political orientation and foundation, $F$(4.259, 570.748) = 2.89, $p$ = .019, $\eta_p^2$ = .02, 95% CI [.00, .04] (see Fig 6).

This interaction effect was slightly stronger same when we controlled for judgment confidence, the agency in IVE, realism in IVE, and VR experience, $F$(4.276, 538.714) = 3.59, $p$ = .006, $\eta_p^2$ = .03 (see the Supplement for more information). Further comparisons showed that for the authority foundation, conservatives conformed less in the presence of AI-controlled avatars ($M$ = 0.31, $SD$ = 0.35) than the human-controlled avatars ($M$ = 0.56, $SD$ = 0.38), $p$ = .047, $d$ = -0.36, 95% CI [-1.35, -0.01]. Other comparisons at this level were nonsignificant.

In the presence of AI-controlled avatars, liberals conformed less when the care was violated ($M$ = 0.23, $SD$ = 0.23) than fairness ($M$ = 0.32, $SD$ = 0.38), $p$ = .028, $d$ = -0.27, 95% CI [-0.60, -0.03], authority ($M$ = 0.35, $SD$ = 0.35), $p$ = .028, $d$ = -0.38, 95% CI [-0.60, -0.03], and loyalty ($M$ = 0.35, $SD$ = 0.31), $p$ = .012, $d$ = -0.32, 95% CI [-0.70, -0.08]. In the presence of human-controlled avatars, liberals conformed less when care was violated ($M$ = 0.25, $SD$ = 0.27) than fairness ($M$ = 0.36, $SD$ = 0.25), $p$ = .038, $d$ = -0.36, 95% CI [-0.57, -0.02], and loyalty ($M$ = 0.38, $SD$ = 0.35), $p$ = .033, $d$ = -0.44, 95% CI [-0.58, -0.02]. In contrast, conservatives in the presence of AI-controlled avatars conformed to all moral foundations to the same extent, $ps$ = .365. In the presence of human-controlled avatars, conservatives conformed less when sanctity was violated ($M$ = 0.19, $SD$ = 0.27) than care ($M$ = 0.36, $SD$ = 0.31), $p$ = .048, $d$ = 0.33, 95% CI [0.004, 0.99]. Remaining comparisons at this level were nonsignificant.

## Discussion

Corroborating the results of Study 1, we found that people changed their private moral character judgments under the pressure of avatars. Confirming our second hypothesis, we did not find a difference in participants' compliance levels between the AI and human-controlled avatars. Although we found that political orientation and moral foundations impacted moral conformity, we cannot conclude if our third hypothesis was confirmed. Liberals conformed less when the care foundation was transgressed but only in comparison to fairness, authority, and loyalty and when avatars were AI-controlled. In contrast, conservatives conformed less when the sanctity than care foundation was broken but only when avatars were human-controlled. These results suggest that moral conformity in immersive virtual environments could be impacted by political orientation.

## General discussion

We investigated whether human and nonhuman sources of social pressure shape moral character inferences. In Study 1, we found evidence that participants' private moral character judgments changed 43% of the time when a group of humans publicly made opposite moral character judgments. In Study 2, in the immersive virtual environment, participants changed their private moral character judgments 30% of the time when the group was represented by avatars controlled by humans and 26% when AI-controlled avatars. However, the difference between human and AI-controlled avatars was not statistically significant.

We also showed that participants' compliance with the group depended on which moral foundation the judged target violated. Under human group pressure in Study 1, participants were less willing to change their moral character judgments when the target person harmed

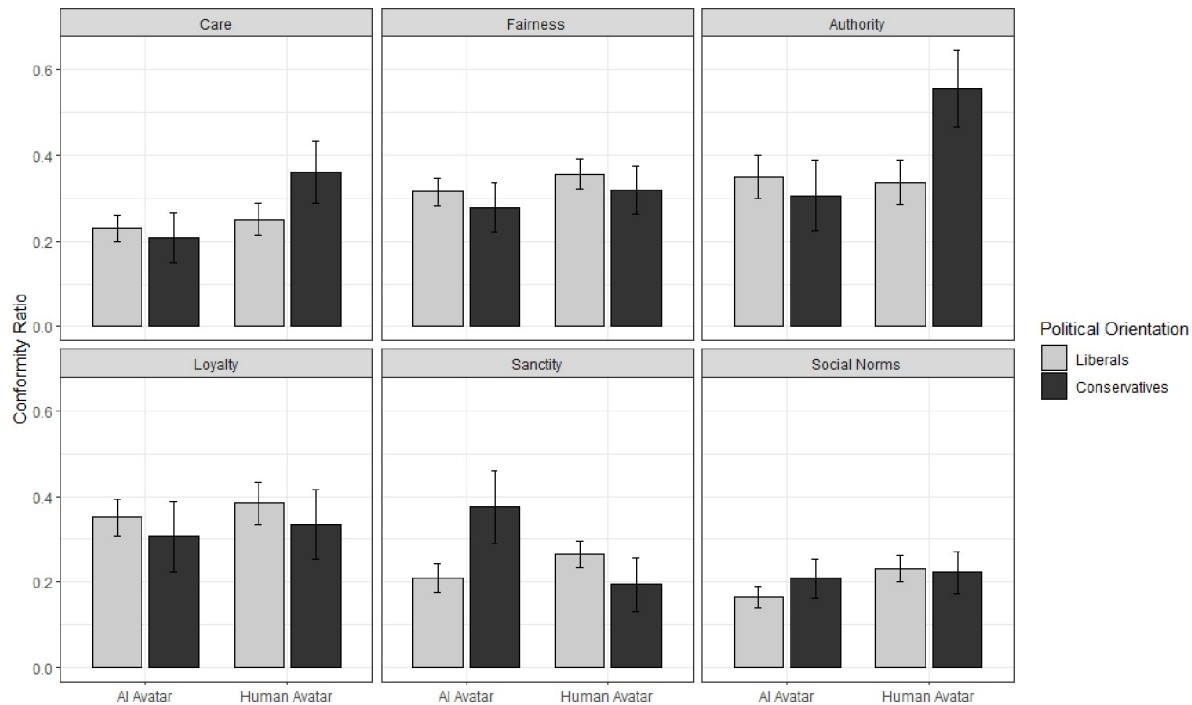

**Fig 6. Mean conformity ratio as a function of the source of social pressure, political orientation, moral domains, and social norms.** The error bars represent one standard error.

others than other moral foundations. In Study 2, this pattern of results was more nuanced since moral conformity depended on political orientation and moral foundations. Nevertheless, in both studies, we found evidence which suggests that moral conformity might vary between moral foundations.

By examining how different sources of social pressure impact moral conformity, we build on and extend the past work in this area. First, previous research found that people comply with the group when judging both immoral and moral actions [9], moral norms [10], when the group is represented by deontologists [12], and when group answers during a video meeting [15], we demonstrated that groups influence how people judged others' moral character. Therefore, the current study demonstrates that conformity arises in person-centred moral judgments [see 17], which may help build a cumulative science of conformity. Second, we substantially extended past research on social conformity in the immersive virtual environment [4, 31] by showing that people's moral character judgements stay under the influence of non-human agents (avatars) independently of whether other human or AI controls them.

Finally, we demonstrated that the moral foundation in which an individual commits transgression impacts the effects of moral conformity. Specifically, we showed that people conform less under group pressure when their moral character judgements concern harm and more when they concern other moral foundations such as fairness, authority, loyalty, or purity. However, this was true only when other humans represented the group. In the immersive virtual reality, where avatars represented the group, we found that moral foundations and political orientation impacted moral conformity.

Therefore, on the one hand, our research provides evidence for the Theory of Dyadic Morality [40], which argues that harm is central to moral cognition and, therefore, equally important for liberals and conservatives. On the other hand, we also found evidence

supporting the Moral Foundations Theory [38], which argues that liberals and conservatives rely on different moral foundations when making moral judgments. Future research would do well to test which of these two theories better explains moral conformity in physical and digital settings.

## Sources of social peer pressure: Human versus AI

For explanatory purposes, we tested the differences in moral conformity between Studies 1 and 2. We believe that the difference in moral conformity between group pressure produced by real humans and avatars is best explained by the Threshold Model of Social Influence model (TMSI); [4]. Unlike avatars, we may assume that real humans represent a higher agency and behavioural realism. Since high agency and behavioural realism lead to high social presence and high social presence results in more significant social influence, people should comply more and change their moral character judgments more frequently under the pressure of actual humans than avatars allegedly controlled by other humans or artificial intelligence (AI).

We found evidence confirming the TMSI model. Participants conform more in the presence of real humans ($M = 0.43$, $SD = 0.20$) than avatars controlled by other humans ($M = 0.30$, $SD = 0.15$), mean difference = -0.13, 95% CI [-0.19, -0.08], $t(167.426) = -4.99$, $p < .001$, $d = -0.74$, 95% CI [-1.05, -0.42] and avatars controlled by AI ($M = 0.26$, $SD = 0.14$), mean difference = -0.17, 95% CI [-0.22, -0.12], $t(167.991) = -6.61$, $p < .001$, $d = -0.97$, 95% CI [-1.29, -0.64]. There was no difference in moral conformity between the presence of human and AI-controlled avatars, mean difference = -0.04, 95% CI [-0.09, 0.01], $t(136) = -1.48$, $p = .142$, $d = -0.25$, 95% CI [-0.59, 0.08]. This lack of difference between the avatar conditions makes even more sense if we look at the measures of agency and realism in the IVE. We did not find any difference between human-controlled and AI-controlled avatars conditions in participants' feelings of presence in the IVE ($M = 3.80$, $SD = 1.08$ vs. $M = 3.64$, $SD = 1.03$, $t(132) = 0.87$, $p = .385$), realism in the IVE, ($M = 2.64$, $SD = 1.01$ vs. $M = 2.46$, $SD = 0.94$, $t(132) = 1.07$, $p = .288$).

The above results set a promising avenue for future research to investigate how social pressure produced by real humans and AI avatars impacts moral conformity. They also confirm a predictor value of the TMSI model and show that it can be used successfully to explain how different virtual and non-virtual peer pressure sources impact people's conformity levels. However, since we did not test all conditions in one study, this explanatory analysis should be treated cautiously. Nevertheless, future studies would do well to test all the sources of peer pressure in one experiment.

Another interesting result from the comparison analysis between Studies 1 and 2 is the effect of political orientation found in the VR but not in the human setting. To our knowledge, there is no reasonable explanation for this difference. The only one that comes to our mind could be related to how political orientation was measured. Maybe the measure we used (participants were asked to report their political ideology on a scale from 1 = *extremely liberal* to 9 = *extremely conservative*) better operationalised political ideology in the UK than in Poland. For example, in contrast to the UK, in Poland, the political landscape is divided between more than two parties, presenting different economic, ideological, and social ideologies. In other words, while in the UK, liberals are liberal regarding the economy, ideology and social issues, in Poland, voters can identify themselves as ideologically liberals but not economically or socially.

## Limitations, implications, and future directions

We acknowledge that our work has certain limitations that might warrant future research. One limitation could be a lack of a control group, as one could argue that some of the moral

character judgments made by participants could naturally change over time. However, past research suggests that moral judgments are generally stable over a short period (e.g., 6–8 days, correlations around 0.6–0.7); [44, 45]. Moreover, participants responded to the question: "Do you think that [TARGET] is mainly a good person or a bad person?" with either the option "Mainly a good person" or "Mainly a bad person". With the Likert scale, variability could be higher as participants could not remember their answers, which is rather unlikely with the current study, where we recorded participants' binary choices. Nevertheless, future studies could control potential variability in moral conformity with the control group.

Another promising avenue for future studies is whether moral conformity observed in our studies was driven by normative or informational influence. One could argue that because of the ambiguity of the dilemmas chosen for the studies, participants relied more on others' judgment regarding informational influence, not normative influence. For example, classic studies on the bystander effect have documented that ambiguity of the emergency may explain why people do not help [46]. Specifically, Latane and Rodin [47] suggest that in an ambiguous situation, each bystander may look to others for guidance before acting and misinterpret others' lack of initial response as a lack of concern.

Some evidence for the informational influence as a mechanism explaining moral conformity could be found in our study. Specifically, we showed that people conformed less when the group pressured them to change their moral judgments about targets who violated a non-moral social norm. Presented violations regarding social norms were likely less ambiguous than violations regarding moral foundations, and as a result, participants were less likely to comply with the group. In contrast, the normative influence as a source of moral conformity in our study is less likely because participants conformed when avatars represented the group. Even if we assume that participants could care if humans who controlled avatars would accept them, it is hard to make the same assumptions about the AI which controlled avatars. Therefore, confidence could explain their compliance that the group knows better how to interpret and judge presented behaviour. Future studies could investigate how people's motivation to be right might impact their moral conformity, as past research found that the importance of the task and incentives impact conformity [48].

Apart from normative and informational influence as potential sources of moral conformity, future research may consider whether people use moral conformity strategically, as research suggests that moral judgments are sometimes motivated by social and personal relationships [49–52]. On the one hand, the relationship regulation theory [53] argues that whether an action would be judged as right or wrong entirely depends on the social-relation context in which it occurs. Therefore, we may assume that people conform more in the presence of friends, family members, or in groups than strangers and out-groups. On the other hand, the dynamic coordination theory [54] argues that people use moral condemnation to decide strategically which side of the conflict they should choose. Therefore, moral conformity may result from an attempt that people make to select the side that seems to have more power in the current social context.

Correspondingly, although people distance themselves from others with different moral convictions [55], we demonstrated that people aligned their moral judgments with those of others when they represented the majority. Thus, our work suggests that moral convictions, which have been theorised to be inflexible and universal [56], are susceptible to intergroup bias. However, as some people resist the majority's influence while others easily align their moral judgments with their peers, future work should examine the role of individual differences in moral conformity. For example, past studies have shown that people with strong moral convictions (vs. weak) against the use of torture resisted a group to more extent [57].

Other research found that participants were less susceptible to a majority's influence when they experienced a fit between their regulator focus and feelings of power [58].

In our study, we observed that individuals did not change their moral judgments as much when presented with a situation where someone violated the care foundation, as opposed to sanctity, loyalty, authority, or fairness violations. We found that this was likely due to the Theory of Dyadic Morality [41], where harm is considered a significant factor in moral cognition, and violations of the care foundation are primarily linked to harming others. However, as we did not measure the perception of harm in this study, we acknowledge this limitation and suggest that future studies address this issue.

Our findings might contribute to understanding how physical and digital groups impact people's moral judgments, providing insight into the interplay of social conformity and moral judgments with important implications for social influence in the modern digital world. For example, past work showed that "bots", identified as white men with many followers, successfully reduced racist slurs on Twitter [59]. Similarly, recent research found evidence that "communicating bots" induce descriptive and prescriptive norms among social networking users and help reduce verbal violence [34]. In both studies, people were unaware that other people in the virtual space were "bots" or, in other words, human profiles controlled by AI.

Our work suggests that the influence of a group of avatars in the immersive virtual environment might have a similar impact on people's moral judgments and behaviour as a group of "bots" in the digital setting. This sets important questions about the consequences of using groups in good or bad faith. More research is needed to establish to what extent groups in the digital setting can influence people's moral cognition and what social consequences this has as we observe the rapid growth of digital communication, which soon, together with social life, may move to different metaverses.

## Conclusion

We demonstrated that human and nonhuman groups changed people's private moral character judgements. More than 43% of the time, participants aligned their moral character judgements with humans in a physical setting and 28% of the time with avatars in the immersive virtual environment. We also found that people's compliance was lower when moral character judgments concerned targets who violated the care foundation compared to other moral foundations but only in the presence of other humans. Moreover, participants' political orientation impacted moral conformity only in the immersive virtual environment. The results suggest that moral character judgments, like other social and moral judgments, are vulnerable and susceptible to the pressure of real and virtual groups.

## Supporting information

**S1 File.**
(DOCX)

## Author Contributions

**Conceptualization:** Konrad Bocian, Lazaros Gonidis, Jim A.C. Everett.

**Data curation:** Konrad Bocian, Lazaros Gonidis.

**Formal analysis:** Konrad Bocian, Lazaros Gonidis.

**Funding acquisition:** Konrad Bocian.

**Investigation:** Konrad Bocian.

**Methodology:** Konrad Bocian, Jim A.C. Everett.

**Project administration:** Konrad Bocian.

**Resources:** Konrad Bocian.

**Software:** Lazaros Gonidis.

**Validation:** Konrad Bocian, Lazaros Gonidis.

**Writing – original draft:** Konrad Bocian, Lazaros Gonidis, Jim A.C. Everett.

**Writing – review & editing:** Konrad Bocian, Lazaros Gonidis, Jim A.C. Everett.

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
