## [Decision Letter · Decision Letter 0]

12 Nov 2023

PONE-D-23-19703Moral Conformity in Digital World: Human and nonhuman agents as a source of social pressure for judgments of moral characterPLOS ONE

Dear Dr. Bocian,

Thank you for submitting your manuscript to PLOS ONE. After careful consideration, we feel that it has merit but does not fully meet PLOS ONE’s publication criteria as it currently stands. Therefore, we invite you to submit a revised version of the manuscript that addresses the points raised during the review process.

We look forward to receiving your revised manuscript.

Kind regards,

Shrisha Rao, Ph.D.

Academic Editor

PLOS ONE

Journal Requirements:

"The preparation of this paper was supported by the Polish National Science Centre grant 2018/02/X/HS6/02164 (MINIATURA 2) and by the European Association of Social Psychology Seedcorn grant awarded to Konrad Bocian."

6. Please ensure that you include a title page within your main document. You should list all authors and all affiliations as per our author instructions and clearly indicate the corresponding author.

8. We note that your Supplementary Figure [Physical (Study 1)] includes an image of a participant in the study.

As per the PLOS ONE policy (http://journals.plos.org/plosone/s/submission-guidelines#loc-human-subjects-research) on papers that include identifying, or potentially identifying, information, the individual(s) or parent(s)/guardian(s) must be informed of the terms of the PLOS open-access (CC-BY) license and provide specific permission for publication of these details under the terms of this license. Please download the Consent Form for Publication in a PLOS Journal (http://journals.plos.org/plosone/s/file?id=8ce6/plos-consent-form-english.pdf). The signed consent form should not be submitted with the manuscript, but should be securely filed in the individual's case notes. 

Please amend the methods section and ethics statement of the manuscript to explicitly state that the patient/participant has provided consent for publication: “The individual in this manuscript has given written informed consent (as outlined in PLOS consent form) to publish these case details”. 

**Additional Editor Comments:**

The reviewers are generally positive, but noted significant issues that the authors need to address. Reviewer 1 in particular raises some points where the authors need to clarify and correct as appropriate.

Reviewers' comments:

Reviewer's Responses to Questions

**Comments to the Author**

1. Is the manuscript technically sound, and do the data support the conclusions?

Reviewer #1: Partly

Reviewer #2: Yes

2. Has the statistical analysis been performed appropriately and rigorously? 

Reviewer #1: Yes

Reviewer #2: Yes

3. Have the authors made all data underlying the findings in their manuscript fully available?

Reviewer #1: Yes

Reviewer #2: Yes

4. Is the manuscript presented in an intelligible fashion and written in standard English?

Reviewer #1: Yes

Reviewer #2: Yes

5. Review Comments to the Author

Reviewer #1: The authors present an interesting paper discussing their findings in moral conformity (which could be seen as an extension or specific case of social conformity), and how moral conformity can occur in virtual immersive environments also with artificial agents. The findings of the paper are interesting and they may be meaningful for future research and for the current advances in digital worlds.

The paper is very well written, has a good flow, it is clear and easy to follow. There are, however, some pieces of relevant information missing, as I will describe below.

However, I have some concerns regarding the experimental design and the explanation of the observed results. I would like that the authors address these concerns, complete their manuscript where necessary and provide the missing and illustrating information.

1. My first concern regarding the experimental design has to do with the question asked after the vignettes are shown to the participants: ´Do you think that "TARGET" is mainly a good or a bad person? ´. My area of expertise is not morals, and I would like to know (and have it described in the paper, since the audience/readers will be varied) whether this is a standard question to investigate moral judgments, if there is precedent in using this question, and if not, how is it justified. I find the question could be difficult to answer and I wonder if "good/bad" is not too polarizing, and if there wouldnt be a more nuanced question to ask.

2. Regarding the study 2, I am missing information regarding the environment, the avatars appearance, and how do they behave or how are they controlled. Do you use pre-recorded animations? Are they static? Do they speak? Do their voices sound human? It is difficult from the brief description of the procedure. Images and preferably a link to a vide would be helpful for the potential readers (and I) to get a feeling of the experimental set-up. Is the participant embodied? Do the avatars look different one another?

3. In page 20, you wrote: "how they shold use the controllers to answer the questions". Could you explain this interaction?

4. How was this virtual environment developed? (e.g. Unity, Unreal, other - please explain). Can you provide some description of the computer and headset that were used for this environment?

5. How long does the task take (both in real life and in VR).

6. In your introduction you discuss social presence, however, it seems like you did not measure social presence in your virtual environment or this was not reported nor considered for your analysis.

7. Even though the experimenter leaves the participant alone in the room after they wear the headset, it is unclear whether the conformity experienced by the participant could be do to "mistrust" of actually being alone or being observed by actual humans during the experiment. Have you controlled for this?

8. Regarding your results, do you think or is there literature investigating whether these conformity effects could be due to self-reflection instead of peer pressure?

9. Your results show that in Study 1, participants revised their responses 43% of the time, while in Study 2 (avatars) participants changed responses between 26% and 30% (depending on condition -human vs. AI driven- and this difference not being significant). First, why do you think this difference arises given that what changes apparently in the format is just whether the people are in real life or represented by avatars. Second, are the differences observed between Study1 and Study2 significant? Have you analyzed or what is your rationale regarding cultural differences? (Study 1 was carried out in Poland, while Study 2 in UK). In what language was the study carried out?

10. There seems to be an effect involving political orientation in the virtual set up but not in the real life one. You do not discuss or provide any rationale on why this may be the case. What are your thoughts?

As a small note, there are still some typos here and there in the manuscript.

Reviewer #2: My evaluation of this paper is very positive. The idea of moral conformity in VR is an important question, especially as everything moves more virtually with AI, and the authors did a careful job exploring this question. I think the studies and the samples and the analytic procedures are all suitable and see no red flags that would prevent me from recommending this current paper for publication.

My main quibble is with the theoretical framing of the morality theories. The theory of dyadic morality (TDM)--rather than 'moral dyad theory'--is less about which 'foundation' is most influenced by various practices, but instead argues that all moral judgment hinges on perceptions of harm. That being said, I think the authors are correct that TDM predicts that the care foundation is most obviously/directly tied to perceptions of harm, and therefore you might find the least movement with these judgments when it comes to conformity (H4 in the current paper). But it's hard to really argue one way or the other without obtaining ratings of perceptions of harm. Perhaps conservatives in S2 see much harm in authority violations.

It might be worth looking at the paper of Ochoa (2023) in Poetics to really make sense of this theoretical landscape.

Otherwise, I appreciated this work and think it is a great fusion of social psychology and AI/VR.

6. PLOS authors have the option to publish the peer review history of their article (what does this mean?). If published, this will include your full peer review and any attached files.

Reviewer #1: No

Reviewer #2: No

---

## [Author Response · Author response to Decision Letter 0]

20 Nov 2023

RESPONSES TO ISSUES RAISED BY REVIEWER 1

1. Reviewer #1 wrote: “The authors present an interesting paper discussing their findings in moral conformity (which could be seen as an extension or specific case of social conformity), and how moral conformity can occur in virtual immersive environments also with artificial agents. The findings of the paper are interesting and they may be meaningful for future research and for the current advances in digital worlds.”

Thank you for your kind words.   2. Reviewer 1 wrote: “My first concern regarding the experimental design has to do with the question asked after the vignettes are shown to the participants: ´Do you think that "TARGET" is mainly a good or a bad person? ´. My area of expertise is not morals, and I would like to know (and have it described in the paper, since the audience/readers will be varied) whether this is a standard question to investigate moral judgments, if there is precedent in using this question, and if not, how is it justified. I find the question could be difficult to answer and I wonder if "good/bad" is not too polarizing, and if there wouldnt be a more nuanced question to ask."

Thank you for this question. There is no “standard” question to investigate moral character judgments. Sometimes, it could be a particular scale measuring a phenomenon or a set of items concerning a dimension of social cognition. Research may use the Likert scale, bipolar scale or even open-ended questions. The way the concept is measured, for example, moral character, depends heavily on a methodological approach. We used two binary responses because of the two-stage nature of our experiments. Since we wanted to investigate group pressure, we needed a measure which would allow us to build a social context in which confederates could always answer the opposite of the participants’ answers. Therefore, if a participant's answer in the first stage was: “Mainly a good person”, in the second stage, the confederates' answer was: “Mainly a bad person”. This way, we measured the general conformity ratio by counting how many times out of 20 participants changed their moral judgments (from good to bad or from bad to good) about the target under group pressures compared to their initial private moral judgments.

3. Reviewer 1 wrote: “Regarding the study 2, I am missing information regarding the environment, the avatars appearance, and how do they behave or how are they controlled. Do you use pre-recorded animations? Are they static? Do they speak? Do their voices sound human? It is difficult from the brief description of the procedure. Images and preferably a link to a vide would be helpful for the potential readers (and I) to get a feeling of the experimental set-up. Is the participant embodied? Do the avatars look different one another?”

As requested by the reviewer, we provide more information about Study 2. We have added this information with the images to the supplement on pages 4-5. 

Two female and one female avatar were seated in front of a table, having their hands placed on the table. They are mainly static except for rotating their heads to face other avatars or the participant. This would happen at random between 30 and 120 seconds for a random period between 4 and 10 seconds. Avatars did not speak. Only the “virtual experiment” would speak, prompting each participant to submit a response. The response was not vocal. Instead, the text “good” or “bad” would appear in each “participant’s” corresponding text box on the whiteboard.

 

4. Reviewer 1 wrote: “In page 20, you wrote: "how they shold use the controllers to answer the questions". Could you explain this interaction?”

Participants were using an Xbox controller and instructed to press the left or right trigger to respond “good” or “bad”. Once the “virtual research assistant” asked the participant to respond with either “good person” or “bad person,” the participant would press the corresponding to their choice controller trigger. 

We have added the above information with the controller's image to the supplement on page 6. 

  5. Reviewer 1 wrote: “How was this virtual environment developed? (e.g. Unity, Unreal, other - please explain). Can you provide some description of the computer and headset that were used for this environment?”

The avatars and their head rotation animations were created in Blender version 2.7, and we then passed these as assets in Unity to design the rest of the experimental setting. We ran the experiment in Unity version 2019.2.4f1. We used a high-performance gaming laptop with an Intel i7-9750H processor, 16 GB of RAM and an Nvidia 2070 Max-Q graphics processor. Regarding the virtual reality headset, we have an Oculus Rift, with its sensor placed and calibrated at a distance of 1m from the participant’s seat. 

We have added the above information to the supplement on page 4. 

 6. Reviewer 1 wrote: “How long does the task take (both in real life and in VR).”

It took around 30 minutes.   7. Reviewer 1 wrote: “In your introduction you discuss social presence, however, it seems like you did not measure social presence in your virtual environment or this was not reported nor considered for your analysis.”

We did measure social presence by asking participants about agency and realism in the IVE. We report all the questions and analyses in the Supplement to maintain the flow of the main paper, as we did not find any significant differences between IVE conditions. 

8. Reviewer 1 wrote: “Even though the experimenter leaves the participant alone in the room after they wear the headset, it is unclear whether the conformity experienced by the participant could be do to "mistrust" of actually being alone or being observed by actual humans during the experiment. Have you controlled for this?

We did not, but we do not see a point here as participants were left alone, and none was in the room, and participants could tell they were alone.   9. Reviewer 1 wrote: “Regarding your results, do you think or is there literature investigating whether these conformity effects could be due to self-reflection instead of peer pressure? 

Thank you for this valuable suggestion. However, we are unaware of any literature investigating where conformity effects could be due to self-reflection. 

 10. Reviewer 1 wrote: “Your results show that in Study 1, participants revised their responses 43% of the time, while in Study 2 (avatars), participants changed responses between 26% and 30% (depending on condition -human vs. AI driven- and this difference not being significant). First, why do you think this difference arises given that what changes apparently in the format is just whether the people are in real life or represented by avatars. Second, are the differences observed between Study1 and Study2 significant? Have you analyzed or what is your rationale regarding cultural differences? (Study 1 was carried out in Poland, while Study 2 in UK). In what language was the study carried out?  Thank you for these questions. We believe that the difference in moral conformity between group pressure produced by real humans and avatars is best explained by the Threshold Model of Social Influence model (TMSI; Blascovich, 2002). The TMSI assumes that the more individuals perceive themselves within interpersonal or social environments, the greater the social influence. However, social presence varies as a function of several factors. Two of them are agency and behavioural realism. In the TMSI, the agency is defined as the extent to which individuals perceive virtual others as representing real persons. Therefore, the agency is represented as a continuum, anchored on the low end, where agents are perceived as entirely controlled by non-human means (e.g., cyborgs, autonomous vehicles). On the high end, we have agents completely controlled by real humans (e.g., avatars., drones). Behavioural realism, in turn, refers to the degree to which virtual objects act as they would in the physical world. Similar to the agency, it is represented on the continuum from the low to the high end. The TMSI assumes that social presence should increase if the agency is high. Social presence should also increase if behavioural realism is high (Blascovich, 2002). Thus, we may assume that real humans, unlike avatars, represent a higher agency and behavioural realism. Since high agency and behavioural realism lead to high social presence and high social presence results in more significant social influence, people should comply more and change their moral character judgments more frequently under the pressure of actual humans than avatars allegedly controlled by other humans or artificial intelligence (AI). 

Our studies confirm the TMSI. First, we compared the results of Study 1 and 2. We found that hat participants conform more in the presence of real humans (M = 0.43, SD = 0.20) than avatars controlled by other humans (M = 0.30, SD = 0.15), mean difference = -0.13, 95% CI [-0.19, -0.08], t(167.426) = -4.99, p < .001, d = -0.74, 95% CI [-1.05, -0.42] and avatars controlled by AI (M = 0.26, SD = 0.14), mean difference = -0.17, 95% CI [-0.22, -0.12], t(167.991) = -6.61, p < .001, d = -0.97, 95% CI [-1.29, -0.64]. We also found that there was no difference in moral conformity between the presence of human and AI-controlled avatars, mean difference = -0.04, 95% CI [-0.09, 0.01], t(136) = -1.48, p = .142, d = -0.25, 95% CI [-0.59, 0.08]. This lack of difference between the avatar conditions makes even more sense if we look at the measures of agency and realism in the IVE. We did not find any difference between human-controlled and AI-controlled avatars conditions in participants' feelings of presence in the IVE (M = 3.80, SD = 1.08 vs. M = 3.64, SD = 1.03, t(132) = 0.87, p = .385), realism in the IVE, (M = 2.64, SD = 1.01 vs. M = 2.46, SD = 0.94, t(132) = 1.07, p = .288). 

We thought about the cultural differences, but the only rationale related to conformity which we can think about is the level of collectivism and individualism. Past research has demonstrated that conformity is more significant in collectivistic countries than in individualistic countries (Bond & Smith, 1996). However, a recent comparison of Poland's and the UK’s cultural characteristics suggests that individualistic and collectivistic division no longer applies to these countries. First, a current hierarchical clustering of 130 countries classified Poland and the UK as Western countries (Awad et al., 2018). Second, more recent clustering conducted in 45 countries confirmed that Poland and the UK are Western countries and showed that distance from the US in collectivism was smaller for Poland (0.059) than for the UK (0.075), (Bago et al., 2022). Therefore, it seems that Poland does not qualify as a collectivistic-eastern country but as an individualistic-western country like the UK. 

Studies were carried out in the native language of the participants. 

11. Reviewer 1 wrote: “There seems to be an effect involving political orientation in the virtual set up but not in the real life one. You do not discuss or provide any rationale on why this may be the case. What are your thoughts?" 

We did not provide any rationale for this result because, to our best knowledge, no good one could explain why the effect of political orientation was found in VR but not in real life. The only one that comes to our mind could be related to how political orientation was measured. Maybe the measure we used (participants were asked to report their political ideology on a scale from 1 = extremely liberal to 9 = extremely conservative) better operationalised political ideology in the UK than in Poland. For example, in contrast to the UK, in Poland, the political landscape is divided between more than two parties, which present different ideologies on economic, ideological and social levels. In other words, while in the UK, liberals are liberal regarding the economy, ideology and social issues, in Poland, voters can identify themselves as ideologically liberals but not economically or socially. 

 12. Reviewer 1 wrote: “As a small note, there are still some typos here and there in the manuscript.” 

Thank you. We have read the manuscript carefully and corrected any typos we could find. 

RESPONSES TO ISSUES RAISED BY REVIEWER 2

1. Reviewer 2 wrote: “My evaluation of this paper is very positive. The idea of moral conformity in VR is an important question, especially as everything moves more virtually with AI, and the authors did a careful job exploring this question. I think the studies and the samples and the analytic procedures are all suitable and see no red flags that would prevent me from recommending this current paper for publication.”

Thank you for appreciating our work.   2. Reviewer 2 wrote: “My main quibble is with the theoretical framing of the morality theories. The theory of dyadic morality (TDM)--rather than 'moral dyad theory'--is less about which 'foundation' is most influenced by various practices, but instead argues that all moral judgment hinges on perceptions of harm. That being said, I think the authors are correct that TDM predicts that the care foundation is most obviously/directly tied to perceptions of harm, and therefore you might find the least movement with these judgments when it comes to conformity (H4 in the current paper). But it's hard to really argue one way or the other without obtaining ratings of perceptions of harm. Perhaps conservatives in S2 see much harm in authority violations. It might be worth looking at the paper of Ochoa (2023) in Poetics to really make sense of this theoretical landscape.”

 Thank you for this valuable comment. We added the lack of harm measure as a potential limitation of both studies and an interesting venue for further research. Now on p. 28, we write: 

In our study, we observed that individuals did not change their moral judgments as much when presented with a situation where someone violated the care foundation, as opposed to sanctity, loyalty, authority, or fairness violations. We found this was likely due to the Theory of Dyadic Morality, where harm is considered a significant factor in moral cognition, and violations of the care foundation are primarily linked to harming others. However, as we did not measure the perception of harm in this study, we acknowledge this limitation and suggest that future studies address this issue.   

3. Reviewer 2 wrote: “Otherwise, I appreciated this work and think it is a great fusion of social psychology and AI/VR.”

Thank you for valuing our work and sharing your kind words.

---

## [Decision Letter · Decision Letter 1]

14 Dec 2023

PONE-D-23-19703R1Moral Conformity in Digital World: Human and nonhuman agents as a source of social pressure for judgments of moral characterPLOS ONE

Dear Dr. Bocian,

Thank you for submitting your manuscript to PLOS ONE. After careful consideration, we feel that it has merit but does not fully meet PLOS ONE’s publication criteria as it currently stands. Therefore, we invite you to submit a revised version of the manuscript that addresses the points raised during the review process.

**Dear authors, ****Please provide the review accordingly the reviewer comments**

We look forward to receiving your revised manuscript.

Kind regards,

Sónia Brito-Costa, Ph.D.

Academic Editor

PLOS ONE

Journal Requirements:

Reviewers' comments:

Reviewer's Responses to Questions

**Comments to the Author**

1. If the authors have adequately addressed your comments raised in a previous round of review and you feel that this manuscript is now acceptable for publication, you may indicate that here to bypass the “Comments to the Author” section, enter your conflict of interest statement in the “Confidential to Editor” section, and submit your "Accept" recommendation.

Reviewer #1: (No Response)

Reviewer #2: All comments have been addressed

2. Is the manuscript technically sound, and do the data support the conclusions?

Reviewer #1: Yes

Reviewer #2: Yes

3. Has the statistical analysis been performed appropriately and rigorously? 

Reviewer #1: Yes

Reviewer #2: Yes

4. Have the authors made all data underlying the findings in their manuscript fully available?

Reviewer #1: Yes

Reviewer #2: Yes

5. Is the manuscript presented in an intelligible fashion and written in standard English?

Reviewer #1: Yes

Reviewer #2: No

6. Review Comments to the Author

Reviewer #1: Thank you for your reply. I unfortunately do not find the supplemental material attached. I still have some comments that are not fully addressed.

Regarding the virtual environment, I think it is important to show how does it look like (images, videos, whatever you can provide) to be able to understand what the participants saw. Is this an immersive environment?

Answers to points 10 and 11 are interesting and make sense, however, I do not see any of this reflected in the manuscript. Please do include this rationale as part of the main manuscript so that the reader can fully make sense of your findings.

Also, coming from a methodological perspective, I think it is important to provide detailed information about the materials and methods as part of the main manuscript, not just the supplemental material. Please include figures and the explanation about the virtual environment as part of the main manuscript in the materials (this sub section is missing in Study 2 and is important for replicability).

Reviewer #2: The authors have addressed all my concerns and also--at least to an outside observer--appear to have addressed the other reviewer concerns. I think this paper provides an elegant demonstration of moral conformity and AI.

7. PLOS authors have the option to publish the peer review history of their article (what does this mean?). If published, this will include your full peer review and any attached files.

Reviewer #1: No

Reviewer #2: No

---

## [Author Response · Author response to Decision Letter 1]

18 Dec 2023

RESPONSES TO ISSUES RAISED BY REVIEWER 1

1. Reviewer 1 wrote: “Regarding the virtual environment, I think it is important to show how does it look like (images, videos, whatever you can provide) to be able to understand what the participants saw. Is this an immersive environment?”

  Reviewer 1 further wrote: “Also, coming from a methodological perspective, I think it is important to provide detailed information about the materials and methods as part of the main manuscript, not just the supplemental material. Please include figures and the explanation about the virtual environment as part of the main manuscript in the materials (this sub section is missing in Study 2 and is important for replicability).”

Thank you for this suggestion. Accordingly, we have moved information about Study 2 setting and environment from the supplement to the main manuscript (p. 20-23).

2. Reviewer 1 wrote: “Answers to points 10 and 11 are interesting and make sense, however, I do not see any of this reflected in the manuscript. Please do include this rationale as part of the main manuscript so that the reader can fully make sense of your findings.”

We much appreciate these suggestions. We have added a new section in the General Discussion called: Sources of social peer pressure: Human versus AI (p. 29). This new section addresses all important information from our responses to points 10 and 11 from the previous revision. 

RESPONSES TO ISSUES RAISED BY REVIEWER 2

1. Reviewer 2 wrote: “The authors have addressed all my concerns and also--at least to an outside observer--appear to have addressed the other reviewer concerns. I think this paper provides an elegant demonstration of moral conformity and AI.”

We are more than happy to hear that. Thank you!

---

## [Editor Report · Decision Letter 2]

23 Jan 2024

Moral conformity in a digital world: Human and nonhuman agents as a source of social pressure for judgments of moral character

PONE-D-23-19703R2

Dear Dr. Bocian,

We’re pleased to inform you that your manuscript has been judged scientifically suitable for publication and will be formally accepted for publication once it meets all outstanding technical requirements.

Kind regards,

Sónia Brito-Costa, Ph.D.

Academic Editor

PLOS ONE
---

## [Editor Report · Acceptance letter]

6 Feb 2024

PONE-D-23-19703R2 

PLOS ONE

Dear Dr. Bocian, 

I'm pleased to inform you that your manuscript has been deemed suitable for publication in PLOS ONE. Congratulations! Your manuscript is now being handed over to our production team.

Kind regards, 

on behalf of

Dr. Sónia Brito-Costa 

Academic Editor

PLOS ONE